# First Detection and Genetic Characterization of Felis catus Papillomavirus Type 11, the First *Treisetapapillomavirus* Type to Infect Domestic Cats

**DOI:** 10.3390/ani15101416

**Published:** 2025-05-14

**Authors:** John S. Munday, Adrienne F. French, Louisa Broughton, Xiaoxiao Lin, Sarah D. Bond, Simona Kraberger, Matthew A. Knox

**Affiliations:** 1Pathobiology, School of Veterinary Science, Massey University, Palmerston North 4410, New Zealand; s.d.bond@massey.ac.nz; 2SVS Laboratories, Hamilton 3200, New Zealand; 3Horse and Cat Vet, Carterton 5713, New Zealand; 4Massey Genome Service, Massey University, Palmerston North 4410, New Zealand; x.x.lin@massey.ac.nz; 5The Biodesign Center for Fundamental and Applied Microbiomics, Arizona State University, Tempe, AZ 85287, USA; simona.kraberger@asu.edu; 6Molecular Epidemiology Laboratory, School of Veterinary Science, Massey University, Palmerston North 4410, New Zealand; m.knox@massey.ac.nz

**Keywords:** cat, classification, evolution, FcaPV, novel virus, papillomavirus, skin

## Abstract

A novel papillomavirus type was amplified and fully sequenced from a domestic cat. This is the 11th papillomavirus type amplified from domestic cats, and the papillomavirus was designated Felis catus papillomavirus type 11 (FcaPV11). By comparing the genetic sequence of FcaPV11 to that of other papillomaviruses, it was determined that FcaPV11 is most likely to be classified within the *Treisetapapillomavirus* genus. Papillomaviruses in the genus have previously been identified in a red fox, a Weddell seal, and a caracal, but this is the first time that a Treisetapapillomavirus has been detected in a domestic species. The novel papillomavirus was detected after unusual papillomavirus-induced cell changes were observed in a sample of skin. While there was no evidence that FcaPV11 caused skin disease in the infected cat, the presence of cell changes indicates the papillomavirus can alter normal cell regulation and is therefore a potential cause of skin disease in cats. FcaPV11 was not detected in additional samples of skin from domestic cats.

## 1. Introduction

Papillomaviruses (PVs) are small, circular, double-stranded DNA viruses. These viruses infect most vertebrate species including reptiles, birds, and mammals [1]. With a handful of notable exceptions, PVs are strictly host specific [1]. Papillomaviruses are thought to have co-evolved with their hosts [1]. As PVs are generally well adapted with their hosts, most PV infections within humans and the domestic species are asymptomatic [2,3,4]. However, as PVs replicate by stimulating cell growth and altering cell differentiation [5], PVs can also cause clinically important lesions. These lesions include self-resolving hyperplastic papillomas (warts), as well as neoplasms that can lead to the death of the infected animal [6,7,8,9,10,11,12,13]. The role of PVs in cancer development is best understood in humans, where the ‘high-risk’ alphapapillomaviruses are thought to cause 4–5% of all cancers in people [14,15].

Papillomaviruses are currently classified using the L1 gene sequence [16]. For a PV to be classified as a new type, there must be less than 90% similarity within the L1 sequence. Papillomavirus types within the same genus typically have greater than 60% similarity within the L1 gene sequence [16]. This classification is important as PVs within the same genus often infect closely related species and have the same clinical manifestations [16,17]. Prior to the present study, the PAVE episteme (https://pave.niaid.nih.gov/, accessed on 30 April 2025) reported 10 Felis catus papillomavirus (FcaPV) types that had been fully sequenced. Of the PV types assigned to a genus (FcaPV1, 2, 3, 4, 5, and 10), all were classified into one of three genera (*Lambdapapillomavirus*, *Taupapillomavirus*, and *Dyothetapapillomavirus*).

The *Lambdapapillomavirus* genus contains FcaPV1 [17,18,19,20]. This PV type was first detected within a skin mass and is now thought to cause papillomas and possibly pre-neoplastic lesions within the mouth and skin of cats [19,21]. FcaPV2 is currently the only FcaPV type classified with the *Dyothetapapillomavirus* genus. FcaPV2 was initially detected in a series of pre-neoplastic skin lesions [22,23]. This PV type has subsequently been shown to be a possible cause of both pre-neoplastic and neoplastic lesions of the skin of cats [24,25,26]. It has also been inconsistently detected in feline oral squamous cell carcinomas (SCCs), although the role, if any, of this PV type in feline oral neoplasia is currently uncertain [12]. While not classified into a genus, FcaPV7 is most similar to FcaPV2 and is thus likely within the *Dyothetapapillomavirus* genus. In contrast to FcaPV2, this PV has not been associated with any disease after being detected on the skin of a human, presumably after recent contact with a cat [27]. The *Taupapillomavirus* genus contains the remainder of the classified FcaPV types. This includes FcaPV3, 4, 5, and 10, which have all been associated with skin neoplasia in cats [28,29,30,31,32,33]. FcaPV6 is most similar to other Taupapillomaviruses and is likely to be classified within this genus. This PV type was detected in a skin neoplasm, although this may have been an incidental finding [34]. No information about disease association is available for FcaPV8 or 9. Cats can also be infected by bovine papillomavirus type 14, which is within the *Deltapapillomavirus* genus [35]. However, this is thought to be a cross-species infection, and it is most likely that BPV14 cannot complete its life cycle in a non-bovine host.

The aim of the present study was to determine the complete genetic sequence of a novel PV type that was detected in a feline histology sample that contained cells with unusual PV-induced cell changes. The amplified genome was most closely related to a PV type detected in a caracal [36] with a likely classification within the *Treisetapapillomavirus* genus. The novel PV type, classified as FcaPV11, was not detected in any of 30 skin samples from 30 additional cats.

## 2. Materials and Methods

### 2.1. Initial Case Summary and Sample Collection

A 1-year-old female spayed Ragdoll cat initially presented to the veterinary clinic due to hair loss and reddening of the skin of the ventrum, under the tail, and around the mouth. The cat was reported to have traumatized the skin, but whether this was due to pruritus or due to behavioral over-grooming was uncertain. The lesions showed only variable and transient response to treatment with antibiotics, corticosteroids, and antihistamines. During the following 2.5 years the lesions were reported to wax and wane, but never completely resolved. The area around the nipple of the right cranial abdominal mammary gland was consistently involved, varying from only mild reddening of the nipple and surrounding skin to an area of ulceration with reddening extending to involve the skin of almost the entire abdomen.

When the cat was 3.5 years old, a biopsy of the inflamed skin surrounding the right cranial abdominal nipple was taken. This revealed multifocal epidermal hyperplasia with superficial pustules, erosion, ulceration, and superficial crusting, accompanied by dermal inflammation with prominent mast cells and eosinophils (Figure 1). These lesions were considered consistent with a diagnosis of a hypersensitivity dermatitis. However, examination of epidermis away from the inflammatory changes revealed areas in which the epidermis was only mildly thickened and covered by mildly increased quantities of loose orthokeratosis (Figure 2). Within the areas of mild epidermal thickening, a small proportion of cells contained large deeply amphophilic cytoplasmic bodies. In some cells, these bodies had expanded to fill the cytoplasm and resulted in displacement of the nuclei to the periphery of the cell. The cytoplasmic bodies were often surrounded by a clear halo (Figure 3). Enlarged nuclei were visible in some of the cells that contained cytoplasmic bodies. The presence of cytoplasmic bodies was not associated with cell necrosis or an inflammatory reaction. The location and appearance of the bodies was considered most consistent with PV-induced cell changes. However, the bodies were larger and more deeply amphophilic than has previously been described due to infections by FcaPV types [37].

### 2.2. Initial PCR and DNA Sequencing

DNA was extracted using a NucleoSpin DNA FFPE XS kit (Macherey-Nagel, Düren, Germany) from a shaving of the formalin-fixed, paraffin-embedded block containing the skin biopsy sample that was noted to contain unusual PV-induced cell changes. Papillomaviral DNA was amplified using the CP4/5 consensus primers as previously described [38]. DNA extracted from a feline cutaneous viral plaque that contained FcaPV3 was used as a positive control for the reactions while no template DNA was added to the negative controls. Amplified PV DNA was purified by incubating the agarose gel in elution buffer overnight at 4 °C and directly sequenced using an ABI3730 Genetic Analyzer (Applied Biosystems Inc., Foster City, CA, USA). The amplified DNA sequences were compared to other sequences in GenBank using the BLAST tool (https://blast.ncbi.nlm.nih.gov/Blast.cgi, accessed on 1 November 2024) [39].

### 2.3. Complete Genome Sequencing of the Novel PV

In response to the results of the diagnostic biopsy, the cat was started on a hypoallergenic diet as well as 1 mg/kg q24 hrs PO clomipramine hydrochloride (Clomicalm, Elanco Animal Health, Auckland, New Zealand) to reduce anxiety. The skin lesions rapidly responded to therapy and only mild reddening was visible when the cat was represented for a recheck one month after the samples had been taken. At this time, to determine if the PV infection had been resolved, a cotton swab was moistened with saline and an approximately 5 cm × 5 cm area of the skin around the nipple was gently swabbed. The cat is currently clinically well with no evidence of skin disease or immunosuppression.

The tip of the swab of the skin was placed in an Eppendorf tube with 1 mL of distilled water and vortexed for 1 min. The tube was then centrifuged at 12,000× *g* for 1 min and the swab tip and the supernatant were removed. DNA was then extracted from pelleted cells within the tube using a NucleoSpin DNA FFPE XS kit as before. The presence of PV DNA was determined by using the CP4/5 PCR primers and the amplified DNA was purified and sequenced as before [38].

To determine the full sequence of the novel PV, ‘outward facing’ primers were designed using Geneious Prime 2019.2.3 based on the short DNA sequence that was amplified by the CP4/5 primers. The primers (FcaPV11InvF 5′-ACTTTTTAACCAGGCAGCAGC and FcaPV11InvR 5′-TGACTGGAAAATTATTGCACAGTTTCT; Integrated DNA Technologies, Coralville, IA, USA) amplified an approximately 7300 base pair (bp) DNA section. Amplification was performed using repliQa HiFi ToughMix (Quantabio, Beverly, MA, USA) according to the manufacturer’s instructions using an Eppendorf Mastercycler Nexus G2 (Hamburg, Germany). An Illumina sequencing library was prepared from the resulting PCR products using the Nextera XT DNA Library Preparation Kit (Illumina Inc., San Diego, CA, USA). Paired-end 2 × 250 bp sequencing of the DNA library was then performed on an Illumina MiSeq sequencer. Around 1.8 million reads were then assembled into a single contiguous sequence using Geneious 10.2.6 [40].

### 2.4. DNA and Protein Sequence Analysis

As the novel PV showed the greatest similarity to Caracal caracal papillomavirus 5 (CcarPV5, GenBank OR915591), the characteristics of the putative viral genes were predicted by comparison to this PV type in Geneious v10.2.6. The presence of conserved protein domains and motifs, and sequences were predicted using the search for motif tool in Geneious and with Interpro (http://www.ebi.ac.uk/interpro/, accessed on 13 March 2025). The predicted novel protein sequences were aligned with corresponding ones from the other four complete Treisetapapillomavirus sequences to check whether conserved protein domains and motifs were conserved within the genus.

### 2.5. Phylogenetic Analysis

Complete genomes of 61 PV types from each of the related genera were obtained from GenBank for comparison with the novel feline PV type. Nucleotide sequences for the L1 ORF were extracted and individually aligned using MAFFT [41] in Geneious v10.2.6 and trimmed using ClipKIT in smart-gap mode, resulting in an alignment of 1602 nucleotides. A maximum likelihood tree was constructed using IQ-TREE Version: 2.2.2.2 [42] with ModelFinder [43], using jmodel test and approximate likelihood-ratio test (aLRT) with 1000 replicates. The resulting tree was produced using the General Time Reversible model with empirical base frequencies, invariable sites, and a gamma distribution with four rate categories (GTR + F + I (0.125) + G4, α = 0.848). Tree visualization, branch concatenation, and annotations were performed with Evolview v3 [44]. Pairwise sequence similarities were calculated from the alignment of the complete L1 ORF of the novel PV with that of other PV types representing other genera.

### 2.6. Nucleotide Sequence Accession Number

The sequence of the novel PV was deposited in GenBank under accession number PV476687.

### 2.7. Detection of DNA Sequences from the Novel PV in Other Feline Skin Samples

DNA was extracted using the above methods from 15 cutaneous SCCs and 15 samples of feline skin without neoplasia. A total of 30 samples were included to give 95% confidence in detecting another case assuming 10% prevalence in the population. These samples were from 30 individual cats. Cats with a variety of skin lesions were included and there was no attempt to preferentially include any particular age of cat, location on the body, or type of skin lesion. None of the samples had been taken specifically for this study, with all taken by clinical veterinarians for histopathological examination as part of a diagnostic work-up to investigate skin disease. The presence of amplifiable DNA in the samples was confirmed by amplifying a section of the feline p53 gene [31]. Specific primers to amplify a 140 bp section of FcaPV11 DNA were designed using Geneious Prime 2019.2.3. DNA was amplified by the primers (FcaPV11F 5′-TTTTTGGGAATGCCAGCGAAG and FcaPV11R 5′-GAATGGGTGGCCAAGTGTTG; Integrated DNA Technologies) using Hot FirePol^®^ Master Mix (Solis BioDyne OÜ, Tartu, Estonia) with an annealing temperature of 60 °C. DNA extracted from the original skin sample containing the unusual cytoplasmic bodies was used as the positive control while no template DNA was added to the negative control.

## 3. Results

### 3.1. Initial PCR and DNA Sequencing

Papillomavirus DNA was amplified by the CP4/5 primes from the sample of skin containing the unusual viral changes and the positive control, but not the negative control reactions. Comparison to other sequences on GenBank revealed that the amplified 312 bp *E1* gene sequence was most similar CcarPV5, with the two sequences being 83.3% similar.

### 3.2. FcaPV11 Complete Gene Sequence

Papillomavirus DNA was amplified from the swab of the skin using the CP4/5 primers. The amplified *E1* ORF sequence was identical to the one initially amplified from the formalin-fixed sample of skin. The outward facing primers amplified an approximately 7300 bp DNA section. Sequencing was performed and the complete genome was determined by assembling the large DNA sequence with the shorter sequence that had been amplified by the CP4/5 primers. The circular DNA contains 7569 bp, with a GC content of 43.8%. The first nucleotide in the ORF *E6* was assigned number 1 in the sequence. As this is the 11th PV type sequenced and classified from domestic cats, it was designated FcaPV11.

### 3.3. Genome Organization of FcaPV11

The PV is predicted to contain seven ORFs that are coded for four early genes (E1, E2, E4, E6, E7) and two late genes (L1, L2; Figure 4). The predicted ORFs and characteristics of their putative protein products are shown in Table 1.

The predicted FcaPV11 E1 protein is 633 amino acids (aa), with N-terminal (aa 2–120) and C-terminal (aa 338–624) ATP-dependent helicase domains and a DNA-binding domain at 195–334. The C-terminal domain contains the conserved ATP-binding site (GPPNTGKS) for the helicase at aa 462–469. The binding of cyclin/cyclin-dependent kinase complexes to E1 is required for the initiation of PV DNA replication [45], and the predicted E1 protein has two cyclin A interaction sites (RXL) at aa 110–112 and 562–564. The FcaPV11 E2 protein is 383 aa in length, consisting of an N-terminal transactivation helicase domain (aa 1–200) and a C-terminal viral DNA-binding domain (aa 303–379). Unlike some other PV types, there is no leucine zipper domain (LX6LX6LX6L) in the putative FcaPV11 E2 protein [46]. The putative FcaPV11 E4 ORF was present within the E2 ORF region, but in a different translation frame with a high leucine content of 14.8%. The putative E6 protein of FcaPV11 consists of 138 aa and contains two conserved zinc-binding domains (CXXC-X29-CXXC) between aa 26 and 62 and aa 98 and 134, but it did not have a PDZ-binding motif (ETQL) in its C-terminus. The 95 aa E7 protein also contained one conserved zinc-binding domain between aa 50 and 86 but lacked a retinoblastoma (pRb) protein-binding site (LXCXE). The ATP-binding site, first cyclin A interaction site, and all zinc-binding domains were conserved in all PV types within the *Treisetapapillomavirus* genus.

The late region encodes two viral capsid proteins, L1 (513 aa) and L2 (521 aa). Both proteins contain a high proportion of positively charged residues (K and R) in the C-terminal end. The highly conserved Y-R dipeptide motif is present in L1 at aa 431–432. The predicted L2 protein also contains two conserved N-terminus furin cleavage motifs between aa 7 and 10 and aa 514 and 517 (RXKR motif), as well as six conserved C-terminus L1-binding sites at aa 97–100, 108–111, 111–114, 139–142, 183–186, and 452–455 (PXXP motif). The Y-R dipeptide motif, first N-terminus furin cleavage motif, and last C-terminus L1-binding sites were conserved in all Treisetapapillomavirus types.

The long control region (LCR) encompasses 765 bp (nt 6805–7569) between L1 and E6. The LCR contains a putative E2-binding site (E2BS), with a consensus sequence ACCN6GGT at nt 7313, and a putative E1-binding site (TTTGCTGTT) at nt 7353.

### 3.4. Phylogenetic Analysis of FcaPV11

Phylogenetic analysis showed that FcaPV11 clustered among *L1* ORF sequences from the *Treisetapapillomavirus* genus (Figure 5). The closest sequence match was to CcarPV5 at 79.2% similarity, while matches with other sequences in the *Treisetapapillomavirus* genus were less similar (60–66.1%) (Table 2). Sequence similarity to representatives from other genera (Gamma, Pi and Taupapillomaviruses were 57.4–64% (Table 2). In comparison to other domestic cat PV types, the *L1* ORF sequence of FcaPV11 was 58.7% similar to FcaPV1 (*Lambdapapillomavirus*) and 59.1% similar to FcaPV2 (*Dyothetapapillomavirus*). Of the PV types within the *Taupapillomavirus* genus, FcaPV11 was most similar to FcaPV3 with 63.1% similarity of the *L1* ORF.

In concordance with previous analyses [36], our analysis grouped *Treisetapapillomaviruses* closest to *Gammapapillomaviruses*, with *Pipapillomaviruses* and *Taupapillomaviruses* placed as other closely related genera.

### 3.5. Detection of FcaPV11 in Additional Skin Samples

While the specific primers amplified FcaPV11 DNA from the original sample, no PV DNA was amplified from any of the 30 additional skin samples.

## 4. Discussion

The novel PV is the 11th PV type thought to have domestic cats as definitive hosts. The novel PV will most likely be classified within the *Treisetapapillomavirus* genus. As no Treisetapapillomaviruses have previously been detected in domestic cats (https://pave.niaid.nih.gov/, accessed on 30 April 2025), this expands the number of PV genera known to have this species as a definitive host. Evidence from other PV types suggests that this novel PV type will only infect cats and is unlikely to have any zoonotic potential [16].

The cat that was infected by FcaPV11 showed skin disease that was histologically consistent with hypersensitivity dermatitis [47], potentially complicated by behavioral over-grooming. The skin lesions are considered unlikely to have been due to the PV infection. Evidence that FcaPV11 did not cause the skin disease includes the history of pruritus. While pruritus has been reported due to extensive PV-induced pigmented plaques in dogs [48], this is not a clinical sign typically associated with PV infection [5]. Similarly, to the authors’ knowledge, PVs have not previously been reported to cause eosinophilic, and mastocytic inflammation was visible in sections of skin from this cat. In addition, the histological changes due to PV infection were only visible in areas of the epidermis that did not contain inflammatory changes. Furthermore, FcaPV11 DNA remained detectible in a swab of the skin even after the skin lesions had almost completely resolved. However, a potential role of FcaPV11 in the development of the skin lesions in this cat cannot be definitively excluded. For example, the presence of the PV infection could have increased the local inflammatory reaction, predisposed to an excessive reaction after exposure to an environmental allergen. A potential role of the PV is supported by the recognized ability of some human cutaneous PV types to alter inflammatory pathways [49], although this typically results in a reduced inflammatory response rather than an increased one. Alternatively, it is possible that FcaPV11 did not cause the skin lesions, but changes to immunoregulation in the skin due to hypersensitivity dermatitis allowed increased replication of FcaPV11.

Only mild thickening of the epidermis was visible in the areas that contained the prominent PV-induced cytoplasmic bodies. Due to the subtle epidermal thickening caused by the PV infection, only lesions attributable to hypersensitivity dermatitis were detected upon clinical examination of the cat. The infection by the novel PV would have probably remained undetected had the diagnostic skin biopsy had not been performed.

Current evidence suggests that most PV infections in cats and other species are asymptomatic [2,3,4,15]. While some aspects of the pathogenesis remain unresolved, it is thought that skin disease in cats due to PVs within the *Dyotheta* and *Taupapillomavirus* genera is due to a reduced ability of the body to inhibit PV replication [7,12]. Lesions then develop due to the greater ability of the PVs to promote cell replication and alter cell differentiation [7,12]. Similarly, infection by FcaPV11 may not be uncommon in cats, but as such few viral copies are present, the PV is not detectible using conventional PCR and does not result in either clinical or histological signs of disease. This would suggest that in the present case reduced skin defenses allowed increased viral replication resulted in detectible quantities of viral DNA, PV-induced cell changes, and mild epidermal hyperplasia. While the cause of the reduced host defenses is uncertain, both chronic damage to the skin caused by self-trauma or the previous use of anti-inflammatory treatment could potentially have allowed greater viral replication [50]. It is possible that, if greater loss of skin defenses is present, more rapid FcaPV11 replication could cause clinically detectible skin disease in cats.

The *L1* ORF of the novel PV was most similar to that of CcarPV5. This PV was detected within rectal swabs of two caracals sampled in South Africa [36]. FcaPV11 also showed significant similarity to CcarPV6, which was detected as a co-infection with CcarPV5 in one of the caracals [36]. Both CcarPV5 and 6 are classified within the *Treisetapapillomavirus* genus and, like the presently described domestic cat, there was no evidence of PV-induced disease in either infected caracal [36]. The first PV type classified within the *Treisetapapillomavirus* genus was detected in a sample of healthy skin from a red fox (*Vulpes vulpes*) [51]. A Treisetapapillomavirus type was also detected in a nasal swab of an apparently healthy Weddel seal (*Leptonychotes weddellii*) [52]. The lack of clinical signs of disease in other species infected by Treisetapapillomavirus types adds to the evidence that these PVs rarely, if ever, cause disease in their hosts [36,51,52]. The low pathogenicity is supported by the close relationship of the Treisetapapillomaviruses to the human Gammapapillomaviruses that infect most people, but they only cause disease when the body is unable to mount an immune response against PV infection [53,54].

Papillomaviruses within the same genus typically infect closely related host species [15,16]. Therefore, it may seem surprising that the Treisetapapillomaviruses have been detected in domestic cats, caracals, seals, and foxes. However, all four species are within the Carnivora order. As PVs are thought to be ancient viruses that co-evolved with their hosts [55,56], it is possible that the Treistapapillomaviruses initially infected a common ancestor of both the Feliformia (domestic cats and caracals) and the Caniformia (foxes and seals) prior to these suborders diverging over 50 million years ago [57]. It appears possible that future studies will identify addition PV types within the *Treistapapillomavirus* genus infecting a wider range of Carnivora species, including domestic dogs. Similarly, members of the *Lambdapapillomavirus* and *Taupapillomavirus* genera are also reported in both Felids and Canids [18,30,58,59], suggestive of a common ancestor.

In addition to the initial sample of haired skin, a total of 30 additional samples of neoplastic and normal haired skin from cats were evaluated for the presence of FcaPV11. The novel PV type was not amplified from any of the additional samples. This suggests that either infection by FcaPV11 is rare in cats or that infection can be focal or present at such low levels to be undetectable using conventional PCR. The probability of there being small amounts of FcaPV11 DNA in the samples was increased by the use of shavings from histology blocks. Compared to taking skin swabs, these blocks contain only small numbers of the superficial squamous epithelial cells that are expected to contain the largest amount of viral DNA [60]. As previously discussed, it appears likely that loss of normal skin defenses allowed more rapid detection by FcaPV11 in the initially infected cat and this resulted in there being enough viral DNA within the samples to allow detection. A further possibility is that there may be differences in susceptibility to infection by FcaPV11 between different breeds of cats. This is supported by the breed predisposition to FcaPV2 infection that has been detected in Devon Rex and Sphynx cats [61,62]. The initially infected cat was a Ragdoll and evaluation of further cats of this breed could allow more frequent detection of this PV type. It is also possible that FcaPV11 infection is restricted to certain areas of the body. Such strict tissue trophism is observed with some human papillomavirus types [5,63], and FcaPV11 infection could potentially be restricted to more lightly haired skin from the ventrum. Although this cannot be excluded, studies in cats have suggested FcaPV types do not have a strict tissue trophism, with multiple types detected both on haired skin and within the oral cavity of cats [28,64,65].

The novel virus in this case was detected because of the large PV-induced amphophilic cytoplasmic bodies that were present within the epidermal cells. Currently in cats, FcaPV1, 2, 3, 4, 5, and 10 have been associated with the development of clinical lesions and histological changes within the skin [10,11,12]. Each of these FcaPV types result in characteristic PV-induced cell changes within the affected epidermis [31,37]. However, none of the previously detected FcaPV types cause such large, round intracytoplasmic bodies as in the present case. The presence of PV-induced cell changes is important as this indicates that FcaPV11 can alter normal epidermal regulation and maturation. The ability of FcaPV11 to influence cell regulation provides additional evidence that this PV type may have the potential to cause skin disease in cats.

Immunohistochemistry to detect PV L1 antigen was not performed in this case because it is uncertain that any of the currently commercially available antibodies would cross-react with the FcaPV11 L1 protein. Therefore, without an appropriate positive control, it would not be possible to interpret an absence of immunostaining within the sample. Immunostaining to detect increased p16^CDKN2A^ protein could have been used to provide further evidence that the FcaPV11 infection was influencing cell regulation by degrading the retinoblastoma protein [66,67]. However, as the PV infection in this case appears most likely an incidental finding and not a cause of disease, this was not performed. Due to the presence of PV-induced cell changes, the expression of PV proteins appears likely to have been present within the sample; however, the presence of PV protein expression within the skin was not evaluated.

The PV-induced cell changes within the epidermis were prominent and were the reason that the case was investigated further. While it is possible that these changes could have been missed by some pathologists when examining histological sections, it appears likely that such prominent changes would most often be detected during routine histological examination. Therefore, as these changes have not been previously reported in a sample of feline skin, this suggests they are infrequently present. This adds further evidence that infection by FcaPV11 is either rare in cats or that infection by FcaPV11 rarely results in changes to epidermal cells.

## 5. Conclusions

This is the first report of FcaPV11. This novel PV type is likely the first member of the *Treisetapapillomavirus* genus to infect domestic cats. Whether or not FcaPV11 can cause disease in cats is uncertain. However, the presence of cell changes and epidermal hyperplasia confirm the virus can alter normal cell regulation, and FcaPV11 may therefore have the potential to cause disease in cats. The presence of other Treisetapapillomaviruses in more diverse animal species may suggest these PVs initially infected a common carnivora ancestor.

## Figures and Tables

**Figure 1 animals-15-01416-f001:**
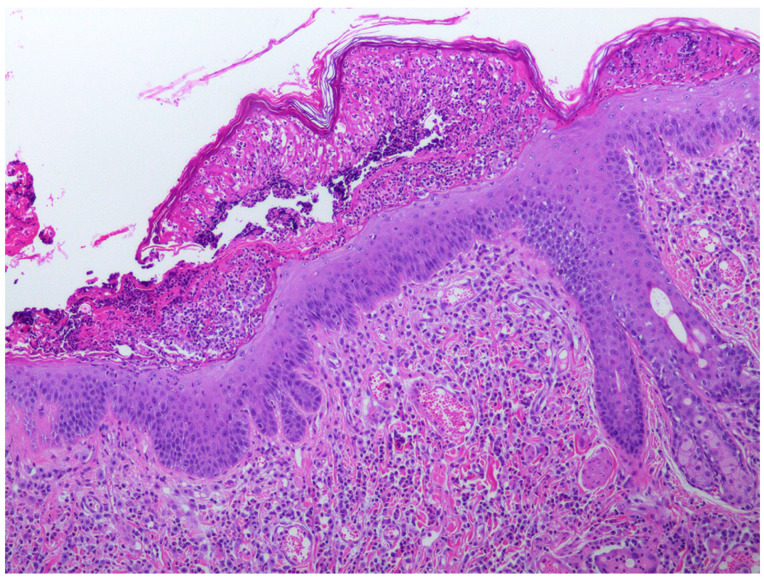
Photomicrograph of the skin sample from around the nipple. The epidermis is thickened and contains a superficial pustule that contains predominantly eosinophils with fibrin, cell debris, and smaller numbers of neutrophils. Large numbers of eosinophils and mast cells are visible within the superficial dermis. Cells containing evidence of papillomavirus infection were not visible within areas of the epidermis showing these inflammatory changes.

**Figure 2 animals-15-01416-f002:**
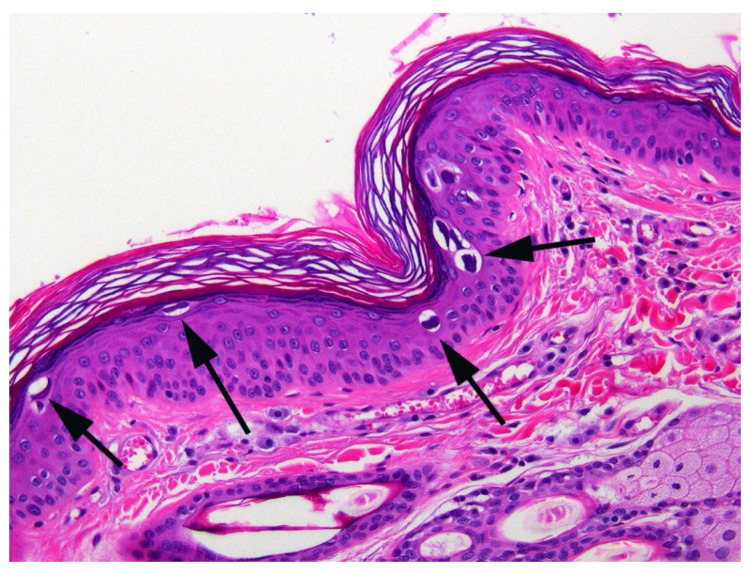
Photomicrograph of the skin sample from around the nipple. The epidermis is mildly thickened and covered by mildly increased quantities of keratin. Large amphophilic cytoplasmic bodies are prominent within the epidermis (arrows).

**Figure 3 animals-15-01416-f003:**
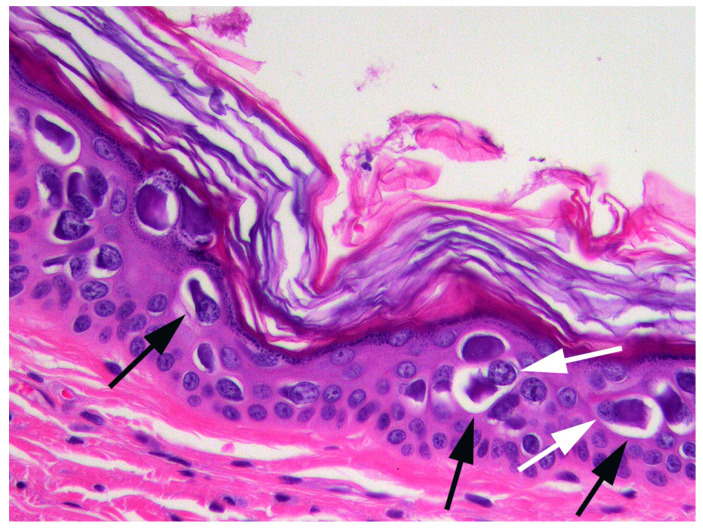
Photomicrograph of the skin sample from around the nipple. The cell changes consist of a large amphophilic, roughly spherical cytoplasmic body that is often surrounded by a clear halo (black arrows). Nuclei are displaced to the periphery of the cell (white arrows).

**Figure 4 animals-15-01416-f004:**
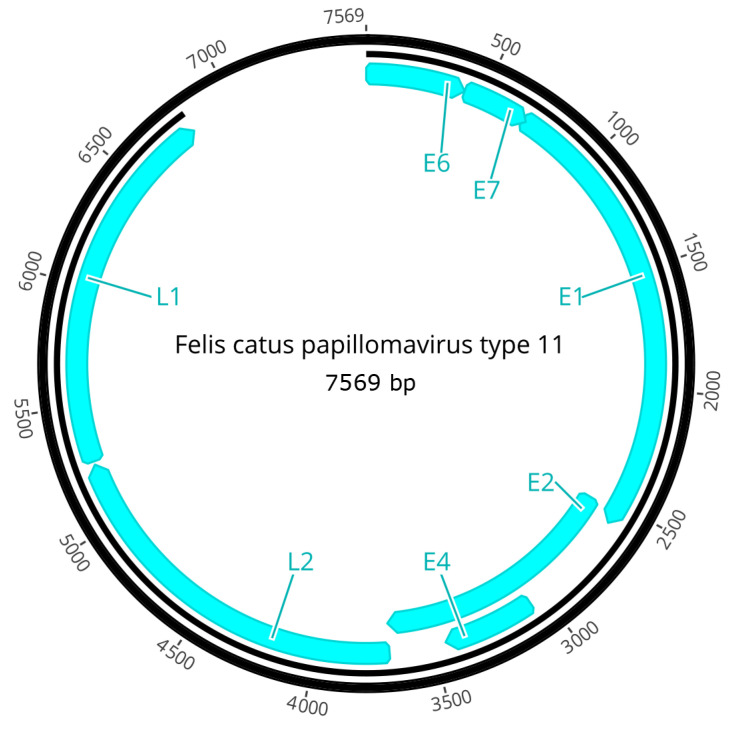
Schematic genomic organization of FcaPV11.

**Figure 5 animals-15-01416-f005:**
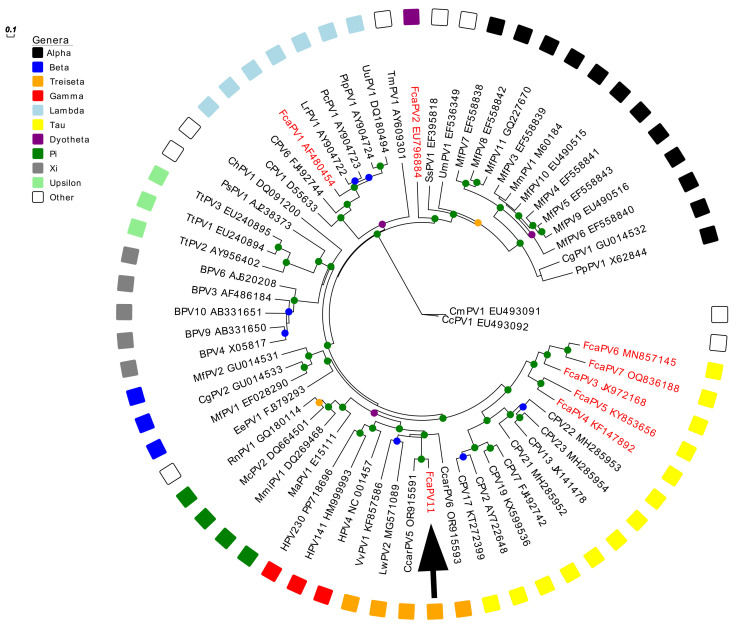
Unrooted maximum likelihood phylogeny based on concatenated nucleotide alignment of the FcaPV11 *L1* ORF sequence (arrow) with 61 other papillomavirus (PV) types of different species and genera. Accession numbers for sequences used are included. Other Felis catus papillomavirus (FcaPV) types are indicated in red. Abbreviations used include Canine papillomavirus, CPV; Caracal caracal papillomavirus, CcarPV; Leptonychotes weddellii papillomavirus, LwPV; Vulpus vulpus papillomavirus, VvPV; Human papillomavirus, HPV; Mesocricetus auratus papillomavirus, MaPV; Micromys minutus papillomavirus, MmiPV; Mastomys coucha papillomavirus, McPV; Rattus norvegicus papillomavirus, RnPV; Erinaceus europaeus papillomavirus, EePV; Macaca fascicularis papillomavirus, MfPV; Colobus guereza papillomavirus, CgPV; Bovine papillomavirus, BPV; Tursiops truncatus papillomavirus, TtPV; Phocoena spinipinnis papillomavirus, PsPV; Capra hircus papillomavirus, ChPV; Lynx rufus papillomavirus, LrPV; Puma concolor papillomavirus, PcPV; Procyon lotor papillomavirus, PlPV; Uncia uncia papillomavirus, UuPV; Trichechus manatus latirostris papillomavirus, TmPV; Sus scrofa papillomavirus. SsPV; Ursus maritimus papillomavirus, UmPV; Macaca mulata papillomavirus, MmPV; Pan paniscus papillomavirus, PpPV. Internal branches are colored based on inferred bootstrap support values, as determined using an approximate likelihood-ratio test (aLRT) with the Shimodaira–Hasegawa-like procedure. Scale bar indicates genetic distance (nucleotide substitutions per site).

**Table 1 animals-15-01416-t001:** Predicted ORFs in the FcaPV11 genome. pI indicates the isoelectric point.

ORF	ORF Location	Length (nt)	Length (aa)	Molecular Mass (kDa)	pI
E1	693–2594	1902	633	70.87	4.86
E2	2536–3687	1152	383	43.84	8.68
E4	3062–3448	387	128	14.09	6.52
E6	1–417	417	138	15.62	7.62
E7	419–706	288	95	10.73	4.49
L1	5263–6804	1542	513	58.33	6.97
L2	3687–5252	1566	521	55.95	4.65

**Table 2 animals-15-01416-t002:** Percent identity between the proposed FcaPV11 and other papillomaviruses based on the pairwise nucleotide alignments of the papillomavirus *ORF L1*. The alignments were performed in Geneious v10.2.6.

Papillomavirus	Host Species	Classification	L1 Similarity (%)
Caracal caracal papillomavirus 5 (OR915591)	Caracal	Treisetapapillomavirus	79.2
Caracal caracal papillomavirus 6 (OR915593)	Caracal	Treisetapapillomavirus	66.1
Micromys minutus papillomavirus 1 (DQ269468)	Eurasian harvest mouse	Pipapillomavirus	64
Vulpes vulpes papillomavirus 1 (KF857586)	Red fox	Treisetapapillomavirus	63.4
Felis catus papillomavirus 3 (JX972168)	Domestic cat	Taupapillomavirus	63.1
Leptonychotes weddellii papillomavirus 2 (MG571089)	Weddell seal	Treisetapapillomavirus	62.3
Felis catus papillomavirus 4 (KF147892)	Domestic cat	Taupapillomavirus	61.3
Human papillomavirus 230 (PP718696)	Human	Gammapapillomavirus	61.2
Felis catus papillomavirus 6 (MN857145)	Domestic cat	Taupapillomavirus	60.7
Felis catus papillomavirus 5 (KY853656)	Domestic cat	Taupapillomavirus	60.5
Felis catus papillomavirus 2 (EU796884)	Domestic cat	Dyothetapapillomavirus	59.1
Felis catus papillomavirus 1 (NC004765)	Domestic cat	Lambdapapillomavirus	58.7

## Data Availability

All data related to this study are presented and published here.

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
