# Peer review of "First Detection and Genetic Characterization of Felis catus Papillomavirus Type 11, the First *Treisetapapillomavirus* Type to Infect Domestic Cats"

_animals, 2025, doi:10.3390/ani15101416_

Round 1

Reviewer 1 Report

Comments and Suggestions for Authors

 I reviewed the article entilited:First detection, and genetic characterization, of Felis catus papillomavirus type 11, the first Treisetapapillomavirus type to infect cat.

In summary 

Lines 27 and 28 what author mean by the sentence and it therefore.please correct the grammatical errors.

Abstract 

Lines 34-36 authors stated  The entire 7569bp genome was amplified and sequenced from a skin swab and the novel 34 PV, designated FcaPV11, contained coding regions that were predicted to produce five early proteins and two late one. Please clarify these sentence and rephrased it.

Last paragraph in abstract should be rephrased again 

Keywords must be reorganized with alphabet order first capital letter 

Line 54 authors stated that PVs can also cause clinically important lesions. These lesions include self-resolving hyperplastic papillomas (warts), but also neoplasms  that can progress to result in the death of the infected animal . authors need to add more references new and relevant.

Line 70 The Lambdapapillomavirus genus contains FcaPV1 . authors need to state more references to this line.

Lines 90-95 the last paragraph in introduction should contain the clear objective of the paper.

Materials and methods

Paragraph related to Initial case summary and sample collection should be written in details 

What about skin affection history?

What about the history of animal?

Please mention the inclusion and  exclusion criteria?

Please justify how authors make calculation of sample size.

Lines 108-119 without any references.please add more related references.

In figure 2 please clarify the roughly spherical cytoplasmic body that is often surrounded by shade with an arrow.

Please add references to this paragraph Complete genome sequencing of the novel PV.

Line 160_165 need to be represented in a good manner with rephrasing.

Paragraph of Detection of DNA sequences from the novel PV in other feline skin should be rewritten again with more details and 

Discussion 

Lines 331_350 all those lines without any refrences how come .this paragraph mist contain many relevant references related to the topic either similar or dissimilar to the paragraph with more clear logical discussion.

The same was observed at lines380-418.i cannot make any clear judgment on the discussion part .therefore you must try to rewrite the discussion section totally from the first line . The discussion section must be totally rewritten again.please justify.

Conclusion is ok

Figure no 3 must be enlarged with a bigger size and perfect resolution.

References 

Many references should be added as suggested please take this in your consideration.

Comments on the Quality of English Language

The English language should be reorganized 

Reviewer 2 Report

Comments and Suggestions for Authors

This study reports the discovery and genomic characterization of FcaPV11, a novel papillomavirus (PV) in domestic cats (Felis catus), representing the first known member of the Treisetapapillomavirus genus in felines. The findings expand our understanding of feline PV diversity and suggest evolutionary links to a common Carnivora ancestor. While the clinical significance of FcaPV11 remains uncertain, its detection in a cat with presumptive allergic dermatitis raises intriguing questions about its potential role in feline skin pathology. This is an interesting manuscript, but I have several follwoing concerns:

  1. The cat had "presumptive allergic dermatitis"—could FcaPV11 have exacerbated inflammation or secondary lesions?
  2. Were other dermatopathogens (e.g., Demodex, bacteria) ruled out? Histopathology images would strengthen claims.
  3. FcaPV11 was undetected in 30 cats—could sampling (e.g., site selection, DNA extraction methods) explain this?

  4. Suggest screening oral/genital mucosa, where PVs often persist asymptomatically.
  5. Did immunohistochemistry confirm PV antigen in the intracytoplasmic bodies?

  6. Were E4 or L1 protein expressions detected (markers of active infection)?

  7. How might FcaPV11 influence "cell regulation"? Does it integrate into the host genome or modulate cell-cycle genes (e.g., p53, Rb)?

  8. Compare FcaPV11 to PVs in other Carnivora (e.g., dogs, mustelids)—are there intermediate hosts or recombination events?

  9. Could FcaPV11 have zoonotic potential? Briefly discuss risks.

  10. Clarify functional predictions for the 5 early (E1, E2, E6, E7?) and 2 late (L1, L2) proteins—any atypical domains?

  11. Are splice sites or promoter regions conserved with other Treisetapapillomavirus members?

  12. Please unify the format of references in the article, including the author's name, the case of words in the title of the article, the writing of the name of the journal, and the page number.

Comments on the Quality of English Language

 The English could be improved to more clearly express the research.

Round 2

Reviewer 1 Report

Comments and Suggestions for Authors

The paper is now improved and I now  accept it but please make the refrences style according to the guidelines of journal.thanks alot

Reviewer 2 Report

Comments and Suggestions for Authors

The authors have addressed all my concerns, I recommend accepting it in current form.